# Numerical Study on a Ductile Fracture Model in Pre-Cracked Tension Tests of SUS304L

**DOI:** 10.3390/ma17020276

**Published:** 2024-01-05

**Authors:** Sung-Ju Park, Kangsu Lee, Woongshik Nam, Kookhyun Kim, Byoungjae Park

**Affiliations:** 1Department of Marine Mobility, Tongmyong University, Busan 48520, Republic of Korea; parksj0314@tu.ac.kr; 2Open Grid Laboratory, Busan 48520, Republic of Korea; 3Korea Research Institute of Ships and Ocean Engineering, Daejeon 34103, Republic of Korea; klee@kriso.re.kr; 4LS Cable & System, Gunpo 15845, Republic of Korea; wsnam@lscns.com

**Keywords:** ductile fracture, Hosford–Coulomb model, pre-cracked tension test, stainless steel, fracture prediction

## Abstract

The effectiveness of a ductile fracture model in accurately predicting fracture initiation has been demonstrated. In this study, we concentrate on applying the ductile fracture model to pre-cracked structures constructed from SUS304L stainless steel with experimental and numerical analyses. The Swift hardening law was employed to extend the plastic behavior beyond the onset of necking. Additionally, the Hosford–Coulomb model, integrated with a damaged framework, was utilized to predict ductile fracture behavior, particularly under non-proportional loading conditions. Tension tests were conducted on various specimens designed to illustrate various fracture modes resulting from geometric effects. Numerical analyses were conducted to explore the loading histories, utilizing an optimization process to calibrate fracture model parameters. The proposed fracture model is validated against pre-cracked structures detailed in a reference paper. The results convincingly demonstrate that the fracture model effectively predicts both fracture initiation and propagation in pre-cracked structures.

## 1. Introduction

The prediction of ductile fracture initiation holds a central position in the domains of structural engineering and materials science, exerting profound influence over the safety, dependability, and functionality of engineering components and structures. In the shipbuilding industry, fracture analysis is crucial for assessing the behavior of structures under large deformations caused by extreme loads, such as ship collisions, grounding, and explosions. Ductile fracture, characterized by its gradual deformation and energy-absorbing properties, stands in stark contrast to brittle fracture, which has the potential to culminate in abrupt and catastrophic failures. In numerous engineering applications, the ramifications of structural failure can be extraordinarily severe, encompassing not only substantial economic repercussions but, of even greater consequence, threats to human safety. The ability to predict ductile fracture empowers engineers with the means to evaluate precisely when and under what circumstances a material or component might approach its deformation limits. This, in turn, furnishes invaluable insights for averting catastrophic failures and underpins the foundations of safety within the realm of engineering.

Researchers have dedicated considerable effort over the past few decades to explore and develop various fracture models to elucidate the behavior of ductile fractures. These models can be broadly categorized as either uncoupled or coupled fracture models [1]. Uncoupled models are expressed by the function of critical value in terms of the stress and strain fields. For instance, Rice and Tracey presented a damage model with a hydrostatic stress term grounded in the theory of void development [2]. It posits that, under tensile loading, macroscopic fracture initiation occurs when voids expand and contract until they reach critical void volume friction. The model employs a critical stress triaxiality value to estimate the point at which void coalescence occurs. The modified Mohr–Coulomb (MMC) model was expressed by the stress-based Mohr–Coulomb criterion into stress and strain fields [3,4]. The Hosford–Coulomb (HC) model, developed based on the Hosford yield criteria, was expressed as mixed stress–strain fields [5]. These models find frequent use in addressing ductile fracture-related challenges, owing to their simplicity in parameter calibration [6,7,8,9,10,11,12,13,14]. The Bressan–Williams-Hill (BWH) criterion was developed based on Hill’s localized necking criterion to introduce a stress-based Forming Limit Curve (FLC) that could be used to predict fracture initiation with shell elements [15]. Subsequently, an extended strain-based version of the BWH model was derived [16,17]. Cerick et al. Lou and Huh [18,19,20] introduced a Lou-Huh ductile fracture model for predicting sheet metals, a model subsequently validated by numerous researchers [21,22,23,24]. Park et al. [25] conducted a comparative study to assess the accuracy of different fracture models (Maximum shear stress, Johnson–Cook [26], Lou–Huh [18], modified Mohr–Coulomb [4], Hosford–Coulomb [5]) in predicting fractures in DH36-grade steel. The findings of this paper indicate that recent fracture models (Lou–Huh, modified Mohr–Coulomb, Hosford–Coulomb), which explicitly incorporate stress triaxiality and lode angle parameters, exhibit superior performance.

Pre-existing cracks within a structure can arise due to various factors, including fatigue, manufacturing imperfections, corrosion, and impacts. These pre-cracks pose significant concerns, particularly in critical applications, as they can compromise structural integrity and safety. Understanding the origin and behavior of these pre-cracks is of the utmost importance for structural assessment and risk mitigation. This study explored the utilization of ductile fracture locus models for predicting complex failure modes in pre-cracked structures, utilizing an engineering aluminum alloy specimen with a side notch and pre-crack [27]. Their study demonstrates the effectiveness of the numerical approach, which combines the fracture locus theory and local damage modeling in accurately predicting failure processes using the ductile fracture model. A computational approach was employed, incorporating a unified elastoviscoplastic-damage model, and the presented model was validated through pre-crack tests [28]. A numerical approach was conducted to simulate a ductile fracture in cracked pipes, employing small punch test data and a multi-axial fracture strain energy density model [29].

Park et al. [25] conducted fracture tests and simulations on EH36-grade steel, and the loading history effect was taken into account by assessing the average values of stress triaxiality and the lode angle parameter. Cerik et al. [30] explored ductile crack formation in DH36-grade steel. While there has been substantial research on the fracture behavior of marine steel, there is a noticeable lack of research on materials relevant to shipbuilding and marine equipment, such as hydrogen tanks. This paper focuses on examining the plastic and ductile fracture characteristics of stainless steel SUS304L used for the hydrogen storage tank. In this pursuit, we employed the Swift hardening law to characterize plasticity behavior, which was complemented by adopting the Hosford–Coulomb model augmented with a damaged framework for simulating ductile fracture. To fine-tune and identify the model parameters, we subjected the specimens to quasi-static tests and conducted finite element analysis under various failure modes. Furthermore, we rigorously validated the presented fracture model, particularly in pre-cracked structure scenarios, by undertaking fracture simulations for the double-edge cracked tension test. The validation process involved meticulously comparing the model predictions with the results obtained from numerical simulations.

## 2. Experiment and Plasticity Model

### 2.1. Fracture Experiment

Quasi-static tests were performed to investigate the plasticity characteristics and fracture model parameters of SUS304L. The chemical composition of this material is provided in Table 1. Figure 1 illustrates the dimensions and labels of the uniaxial tension specimens specifically designed to induce different fracture modes. All specimens had a thickness of 2.0 mm. The standard flat bar type specimen (FB) with a 60.0 mm gauge length and 10.0 mm gauge section was specifically designed to obtain mechanical properties in compliance with ASTM E08 standards [31]. The notch tension specimen (NT) featured a circular cut-out with a 20.0 mm radius for the notch. The plane strain tension specimen (PST) had a 4.0 mm radius for the notch. The shear specimen (SH) was specially designed to occur pure shear in the uniaxial tension test. Uniaxial tension tests were conducted using a servo-mechanical loading frame (Samyeon Tech, Daegu, Republic of Korea) with a 50 mm extensometer (Epsilon Tech, Jackson, WY, USA). The testing speed was set at 2.0 mm/min, and all tests were performed at room temperature.

### 2.2. Plasticity Model

The exact flow stress information during plastic deformation is essential to improve the accuracy of fracture simulation. The yield stress criterion $\left(f\right)$ of isotropic material is expressed by the von Mises equivalent stress $\left(\bar{\sigma }\right)$ and deformation resistance $\left(k\right)$:(1)$$f\left[\sigma ,k\right]=\bar{\sigma }-k\left[{\bar{\varepsilon }}_{p}\right]=0$$

The $\bar{\sigma }$ is a scalar value computed from the Cauchy stress tensor, σ. ${\bar{\varepsilon }}_{p}$ is the equivalent plastic strain. An associated flow rule follows the plastic potential theory as shown below:(2)$$d\varepsilon_{p}=d\lambda \frac{\partial f}{\partial \sigma }$$

As shown in Figure 2, plastic deformation occurs when σ exceeds the surface yield. The plastic multiplier dλ>0 is described using the hardening parameter. In this study, the Swift isotropic hardening law was adopted to describe the stress–strain relationship over the onset of necking:(3)$$k\left[{\bar{\varepsilon }}_{p}\right]=A{\left(\varepsilon_{0}+{\bar{\varepsilon }}_{p}\right)}^{n}$$

The Swift isotropic hardening law is defined by three parameters $\left\{\varepsilon_{0},\;A,\;n\right\}$.

### 2.3. Plasticity Model Calibration

Before the onset of necking, the flow stress was obtained from the tensile test for the FB specimen. The hardening model parameters are determined by an iterative approach based on experimental and numerical results for the FB and NT specimens. The stress–strain curves before the onset of necking were obtained using standard tension tests on flat bar specimens, and the flow stress beyond the large plastic strain region was extrapolated using the hardening law. The presented hardening model parameters were verified through numerical analysis for the cases of NT, PST, and SH. The detailed calibration procedure was introduced in a previous study [6]. Figure 3 illustrates the finite element (FE) model, which includes one-eighth of the gauge section and was created using eight-node solid elements (C3D8R). The element size of 0.1 mm was determined through a convergence test, i.e., ten elements were meshed at half thickness. Figure 4 shows the hardening curves of the tension test for FB and extrapolated flow stress using the MATLAB curve fitting tool (Natick, MA, USA) with the Swift hardening equation and the test result. The identified Swift hardening model parameters were $A=1778.0,\varepsilon_{0}=0.1514,n=0.928$, respectively.

## 3. Fracture Model Calibration

### 3.1. Characterization of Stress State

The stress state of an isotropic ductile material can be elucidated through dimensionless parameters as follows: stress triaxiality denoted as η and the lode angle parameter represented as $\bar{\theta }$. These two stress state parameters can be represented in the plane stress state, as shown in Figure 5. These parameters are formulated based on the invariants of the Cauchy stress tensor and can be articulated as follows:(4)$$I_{1}=tr\left[\sigma \right]$$
(5)$$J_{2}=\frac{1}{2}\left(s:s\right)$$
(6)$$J_{3}=det[s]$$

$I_{1}$ represents the first invariant of the Cauchy stress tensor, denoted as σ. $J_{2}$ and $J_{3}$ are the second and third invariants of the deviatoric stress tensor, referred to as s. They represent the combined influence of the magnitude and direction of shear stress. η and $\bar{\theta }$ are expressed as follows:(7)$$\eta =\frac{I_{1}}{3\sqrt{3J_{2}}}$$
(8)$$\bar{\theta }=1-\frac{2}{\pi }\cos^{-1}\left[\frac{3\sqrt{3}}{2}\frac{J_{3}}{{\left(J_{2}\right)}^{\frac{3}{2}}}\;\right]=1-\frac{2}{\pi }\cos^{-1}\left[\frac{27}{2}\frac{J_{3}}{{\bar{\sigma }}^{3}}\;\right]$$

$\bar{\sigma }$ means the equivalent von Mises stress:(9)$$\bar{\sigma }=\sqrt{3J_{3}}=\frac{1}{\sqrt{2}}\sqrt{{\left(\sigma_{1}-\sigma_{2}\right)}^{2}+{\left(\sigma_{2}-\sigma_{3}\right)}^{2}+{\left(\sigma_{3}-\sigma_{1}\right)}^{2}}\;$$

The principal stresses $\left\{\sigma_{1},\sigma_{2},\sigma_{3}\right\}$ can be written as η and $\bar{\theta }$:(10)$$\sigma_{1}=\bar{\sigma }\left(\eta +\frac{2}{3}\cos\left[\frac{\pi }{6}\right]\left(1-\bar{\theta }\right)\right)$$
(11)$$\sigma_{2}=\bar{\sigma }\left(\eta +\frac{2}{3}\cos\left[\frac{\pi }{6}\right]\left(3+\bar{\theta }\right)\right)$$
(12)$$\sigma_{3}=\bar{\sigma }\left(\eta -\frac{2}{3}\cos\left[\frac{\pi }{6}\right]\left(1+\bar{\theta }\right)\right)$$

### 3.2. Hosford–Coulomb Fracture Model

The Hosford–Coulomb (HC) model with the damaged framework is widely used to predict the onset of the fracture initiation of ductile material under non-proportional loading. The Hosford–Coulomb model was represented by the Tresca equivalent stress with the Hosford equivalent stress, $\sigma_{HC}$:(13)$$\sigma_{HC}+c\left(\sigma_{1}+\sigma_{3}\right)=b$$
(14)$$\sigma_{HC}={\left\{\frac{1}{2}\left({\left(\sigma_{1}-\sigma_{2}\right)}^{a}+{\left(\sigma_{2}-\sigma_{3}\right)}^{a}+{\left(\sigma_{1}-\sigma_{3}\right)}^{a}\right)\right\}}^{\frac{1}{a}}$$
where b is the material-dependent constant and c is the friction coefficient. $\sigma_{HC}$ is expressed in terms of the modified Haigh–Westergaard coordinates $\left\{\bar{\sigma },\eta ,\bar{\theta }\right\}$ using Equations (10)–(12):(15)$$\bar{\sigma }=\sigma_{f}\left[\eta ,\bar{\theta }\right]=\frac{b}{{\left\{\frac{1}{2}\left({\left(f_{1}-f_{2}\right)}^{a}+{\left(f_{2}-f_{3}\right)}^{a}+{\left(f_{1}-f_{3}\right)}^{a}\right)\right\}}^{\frac{1}{a}}+c\left(2\eta +f_{1}+f_{3}\right)}$$
where ${\bar{\sigma }}_{f}$ is the von Mises equivalent stress at fracture under proportional loading. The lode angle parameter function $\left\{f_{1},f_{2},f_{3}\right\}$ is expressed as follows:(16)$$f_{1}\left[\bar{\theta }\right]=\frac{2}{3}\cos\left[\frac{\pi }{6}\left(1-\bar{\theta }\right)\right]$$
(17)$$f_{2}\left[\bar{\theta }\right]=\frac{2}{3}\cos\left[\frac{\pi }{6}\left(3+\bar{\theta }\right)\right]$$
(18)$$f_{3}\left[\bar{\theta }\right]=-\frac{2}{3}\cos\left[\frac{\pi }{6}\left(1+\bar{\theta }\right)\right]$$

Derived from the isotropic power law, ${\bar{\varepsilon }}_{p}=k^{-1}\left[\bar{\sigma }\right]$, the fracture strain of the Hosford–Coulomb model for a proportional loading path, ${\bar{\varepsilon }}_{HC}^{pr}$ can be reformulated in a mixed stress/strain space $\left\{\eta ,\bar{\theta },{\bar{\varepsilon }}_{HC}^{pr}\right\}$ as follows:(19)$${\bar{\varepsilon }}_{HC}^{pr}\left[\eta ,\;\bar{\theta }\right]=b{\left(1+c\right)}^{1/n_{HC}}\;\;g\left[\eta ,\;\bar{\theta }\right]$$
(20)$$g\left[\eta ,\;\bar{\theta }\right]={\left\{\frac{1}{2}\left({\left(f_{1}-f_{2}\right)}^{a}+{\left(f_{2}-f_{3}\right)}^{a}+{\left(f_{1}-f_{3}\right)}^{a}\right)\right\}}^{\frac{1}{a}}+c{\left(2\eta +f_{1}+f_{3}\right)}^{\frac{1}{n_{HC}}}$$

The coefficient, $n_{HC}$, is generally set to 0.1 for general ductile steel [18,19]. Finally, the Hosford–Coulomb model consists of three parameters $\left\{a,b,c\right\}$. The damage framework is used to consider the non-proportional loading effect. In this study, the linear damage accumulation law is adopted with the HC model [20]. The damage indicator, $D\in \left[0,1\right]$, is expressed by the equivalent plastic strain, ${\bar{\varepsilon }}_{p,f}$, and the integration of the weighting function, ${\bar{\varepsilon }}_{HC}^{pr}\left[\eta ,\bar{\theta }\right]$ along the loading history, as follows:(21)$$D=\int_{0}^{{\bar{\varepsilon }}_{p,f}}\frac{d{\bar{\varepsilon }}_{p}}{{\bar{\varepsilon }}_{HC}^{pr}\left[\eta ,\;\bar{\theta }\right]}$$

### 3.3. Fracture Model Parameter Identification

The commercial finite element analysis software Abaqus 2023 was used to investigate the history of the stress state to fracture initiation. The accuracy of the numerical simulation was verified by comparing it to the test results, as illustrated in Figure 6. Determining the fracture initiation displacement is challenging, especially if it occurs inside the specimen, even when measured using a monitoring device. Fracture initiation was assumed to take place at the element where the maximum equivalent plastic strain was observed during fracture displacement. Figure 7 displays the contour plots obtained from finite element analysis with equivalent plastic strain. In the notch tension specimen, fracture initiation occurred at the center of the notch, while in the plain strain tension specimen, it was observed at the side of the notch. For the shear specimen, a fracture occurred at the center of the notch. Figure 8 displays the loading history leading to fracture, with the lowest ductility observed in the shear fracture mode.

Optimization was carried out to calibrate the HC model parameters $\left\{a,b,c\right\}$ using the objective function $f\left[\chi \right]$. The linear damage model was used for the optimization procedure described as follows:(22)$$f\left[\chi \right]=\sum_{i=1}^{3}{\left(\int_{0}^{{\bar{\varepsilon }}_{f}}\frac{d{\bar{\varepsilon }}_{p}}{{\bar{\varepsilon }}_{f}^{pr}\left[\eta ,\theta \right]}-1\right)}^{2}$$

Therefore, the optimization problem is expressed as
(23)$$\chi =\underset{\chi }{argmin}\left[f\left[\chi \right]\right]$$

The index i indicates the number of experiments for calibration. In this study, three tests, NT, PST, and SH were used for calibration. The optimization procedure was conducted using MATLAB, and the calibrated HC model parameters were a=1.3358, b=0.8047, c=0.0002. Figure 9a illustrates 3D fracture loci with a red line representing the plane stress condition, while Figure 9b shows the fracture loci under the plane stress condition. The ductility limit of SUS304L was dependent on the lode angle parameter. A change in stress triaxiality did not significantly affect the variation in fracture strain. It is worth noting that the fracture locus exhibited the lowest ductility in the plane strain tension state.

## 4. Application to Pre-Cracked Tension Test

### 4.1. Description of Reference Test

Tension tests were conducted on SUS304L specimens with an initial crack, named the double-edge cracked tension test (DECT) [28]. These experiments were specifically designed to resemble structures with an initial crack. Figure 10 illustrates the specimen’s geometry, with a thickness of 1.2 mm. Furthermore, standard dog bone-type specimens were subjected to tensile tests to acquire the mechanical properties of SUS304L. In all cases, the tests continued until fracture took place at a consistent loading rate of 1 mm/min.

### 4.2. FE Model and Fracture Simulation

To perform fracture simulation using the proposed HC model for SUS304L, a quarter finite element (FE) model of DECT was created. The model used a reduced integration-type solid element (C3D8R in Abaqus). as illustrated in Figure 11. The finite element model was created under 1/8 of the symmetric conditions for length, width, and thickness. The model was meshed with six elements in half of its thickness. The enforced displacement was applied at the top node. Mesh sensitivity issues near the crack tip line were addressed in a previous study, and the element length around the crack tip was determined to be 0.1 mm [22]. The ABAQUS/explicit user-defined subroutine VUMAT was developed to define the damage indicator at every Gauss point in finite element analysis [6]. This element experienced a loss of stiffness when the damage indicator reached a value of one.

Swift model parameters were re-modeled to improve the accuracy of numerical prediction and also address the effect of uncertainty in the material property ranges. As shown in Figure 12, the fracture simulation for DECT was performed with the newly identified Swift model parameters of $A=1517.0,\varepsilon_{0}=0.1156,n=0.6265$. The accuracy of the fracture model was validated by comparing the force and displacement curves between the experiment and fracture simulation.

Figure 13 presents the test results and numerical results from the reference paper [28] alongside the fracture simulation results using the HC model. It is demonstrated that the error in terms of fracture displacement was reduced when utilizing the HC model compared to the viscoplastic-damage model presented in the previous study.

## 5. Conclusions

The research presented here focuses on the plasticity and ductile fracture models applicable to SUS304L steel, employing the Swift isotropic hardening law for plasticity and the Hosford–Coulomb model with a damaged framework for simulating ductile fracture. The results provide the below conclusions:Quasi-static tensile tests were conducted on SUS304L, a material commonly used for structural steel and, more recently, in hydrogen tanks. The flow stress beyond the onset of necking was obtained for numerical analysis involving large deformation and fracture.Numerical simulations were employed to obtain the loading history until the initiation of fracture, and the Hosford–Coulomb model parameters were determined using an optimization process within a damaged framework. Figure 14 illustrates the 2D ductile fracture loci of AH36, DH36, EH36, and SUS304L steel under plane stress conditions. In all fracture models, the lowest ductility was observed in the plane strain tension state. Notably, SUS304L steel exhibited lower ductility compared to other general shipbuilding steels.The proposed Hosford–Coulomb (HC) model underwent validation through numerical simulations of double-edge cracked tension tests. These results underscore the effectiveness of the HC model in reducing errors in fracture displacement predictions when compared to previous methods, thereby enhancing the accuracy of fracture simulations.

This work contributes to the understanding and prediction of ductile fracture behavior in pre-cracked structures, providing valuable insights for structural assessment, risk mitigation, and the development of safer engineering components and structures.

## Figures and Tables

**Figure 1 materials-17-00276-f001:**
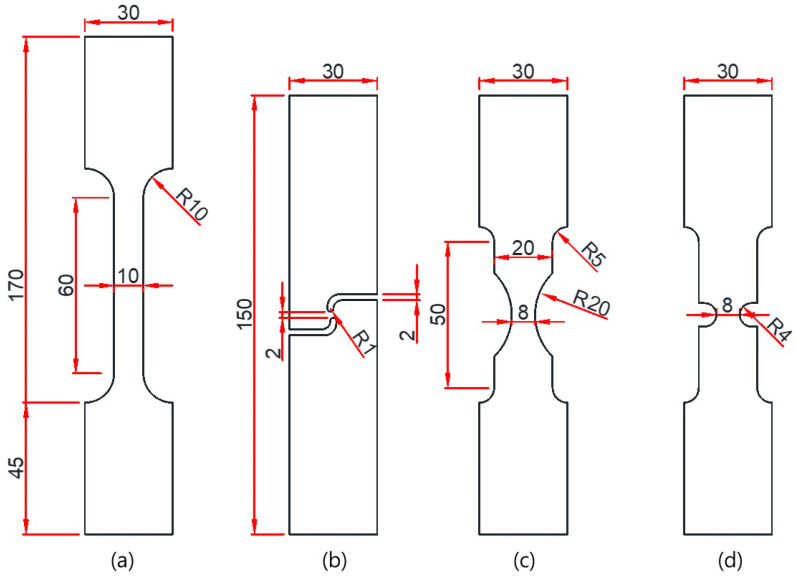
Design of specimens. (**a**) FB; (**b**) SH; (**c**) NT; (**d**) PST.

**Figure 2 materials-17-00276-f002:**
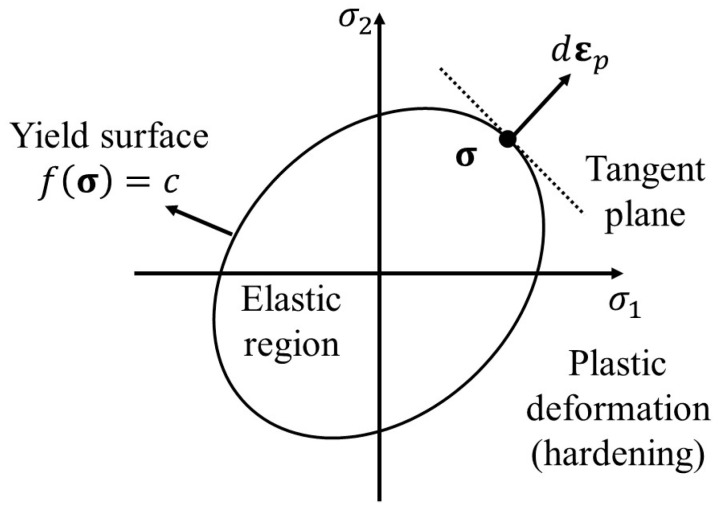
Associated flow rule in principle stress space.

**Figure 3 materials-17-00276-f003:**
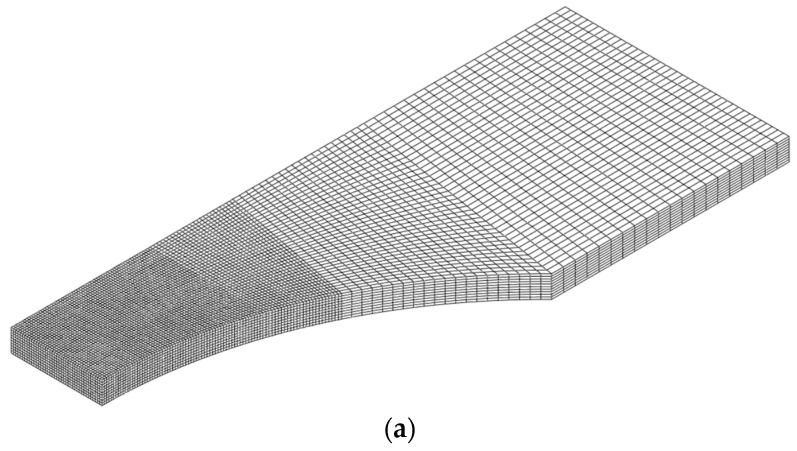
FE models; (**a**) NT; (**b**) PST; (**c**) SH.

**Figure 4 materials-17-00276-f004:**
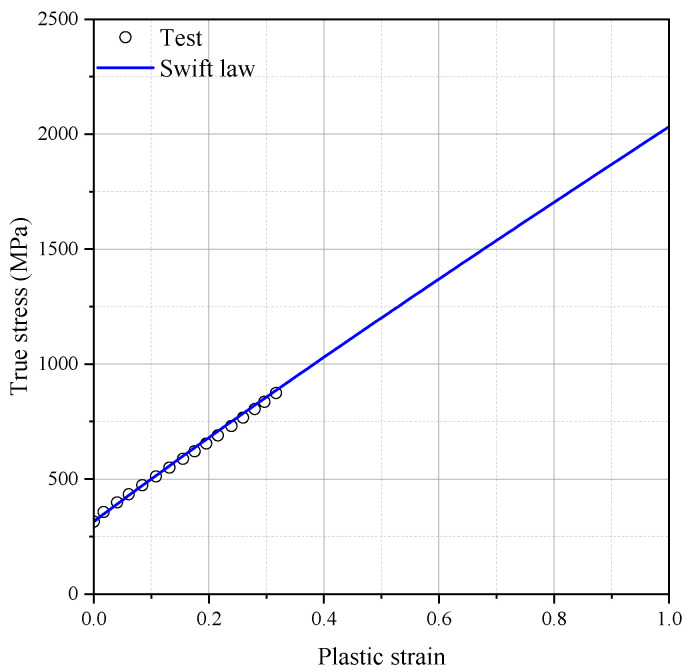
Uniform true stress and uniform true strain curves of SUS304L.

**Figure 5 materials-17-00276-f005:**
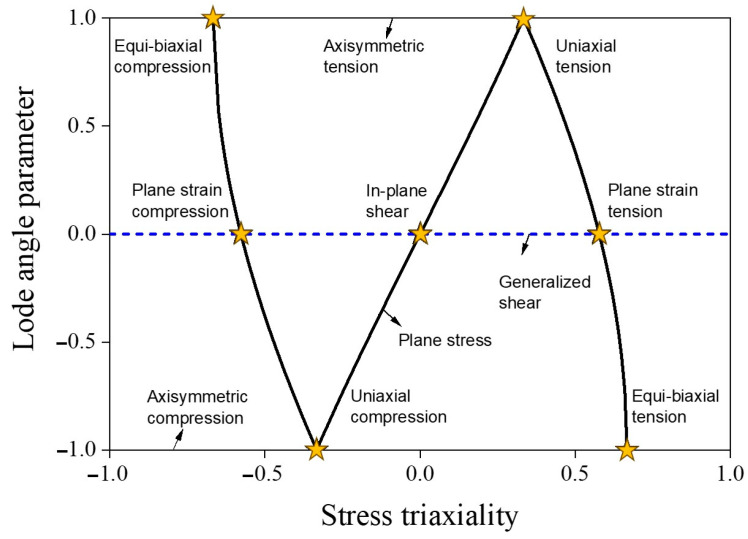
Stress state parameters in the plane stress state [32].

**Figure 6 materials-17-00276-f006:**
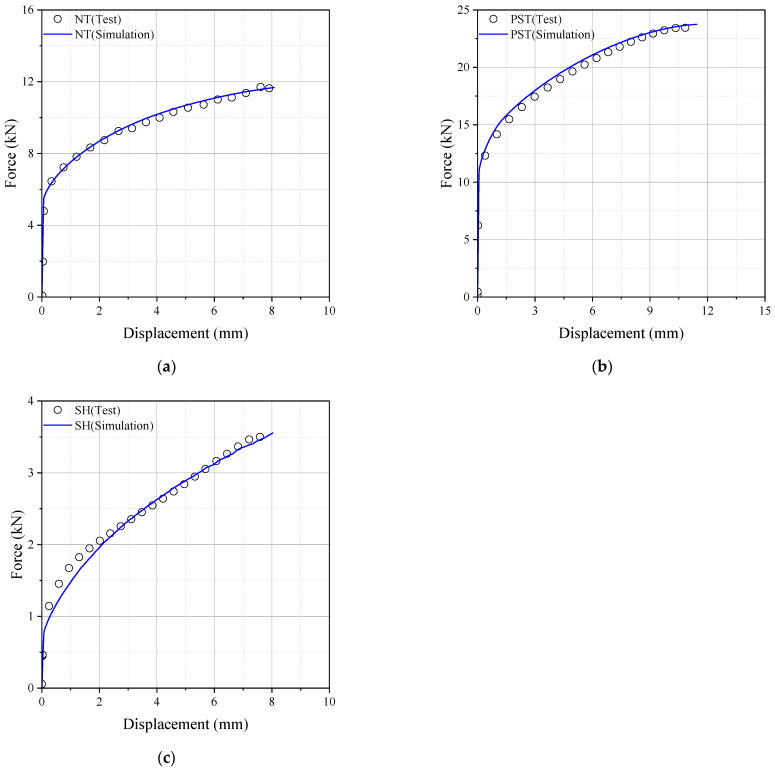
Force–displacement curves (**a**) NT; (**b**) PST; (**c**) SH.

**Figure 7 materials-17-00276-f007:**
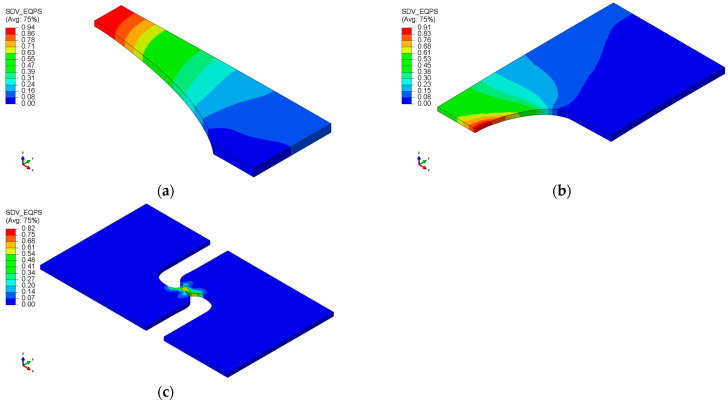
Contour plots of equivalent plastic strain (**a**) NT; (**b**) PST; (**c**) SH.

**Figure 8 materials-17-00276-f008:**
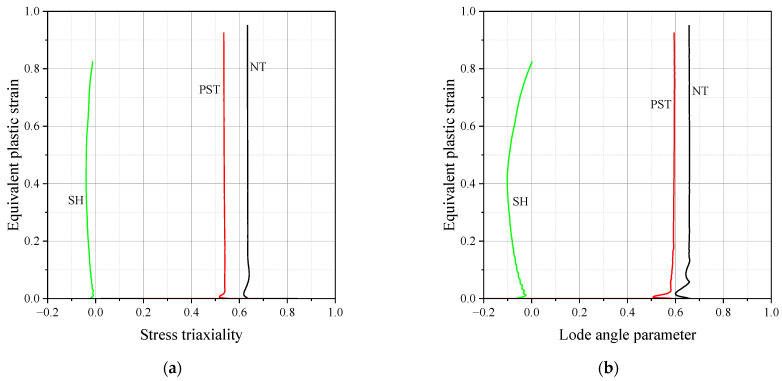
The loading histories to fracture (**a**) stress triaxiality; (**b**) the lode angle parameter.

**Figure 9 materials-17-00276-f009:**
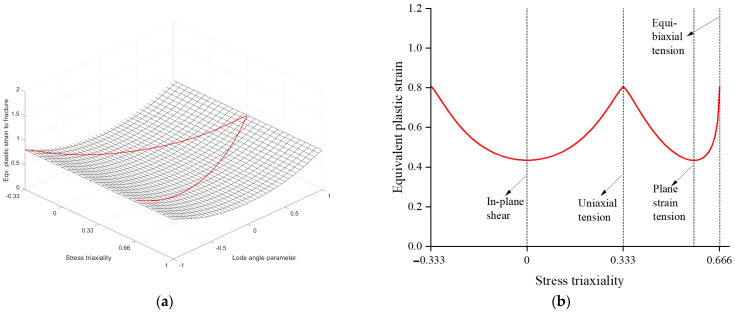
(**a**) depicts 3D fracture loci with a red line representing the plane stress condition, while (**b**) specifically displays the fracture loci under the plane stress condition.

**Figure 10 materials-17-00276-f010:**
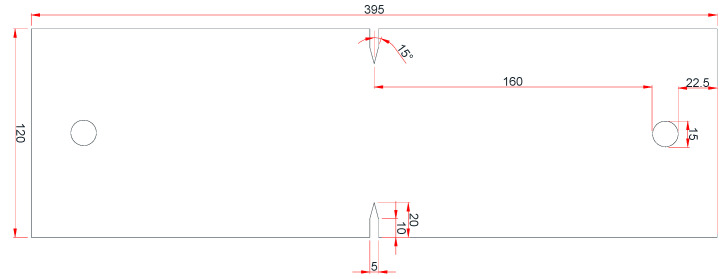
Design of DECT specimen.

**Figure 11 materials-17-00276-f011:**
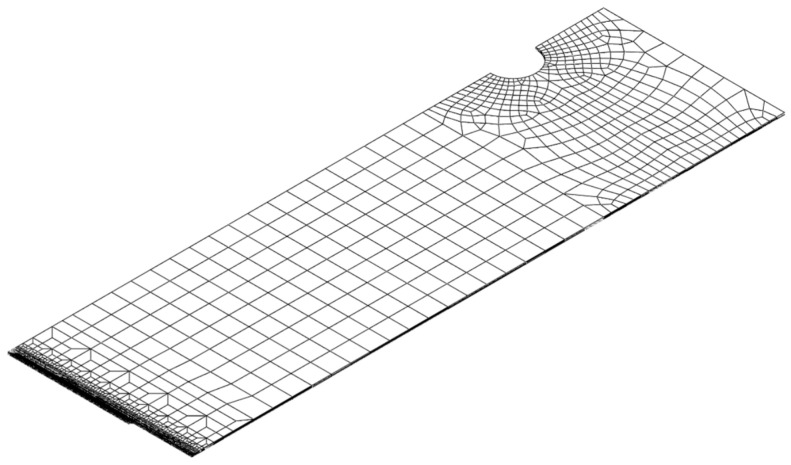
FE model of DECT specimen.

**Figure 12 materials-17-00276-f012:**
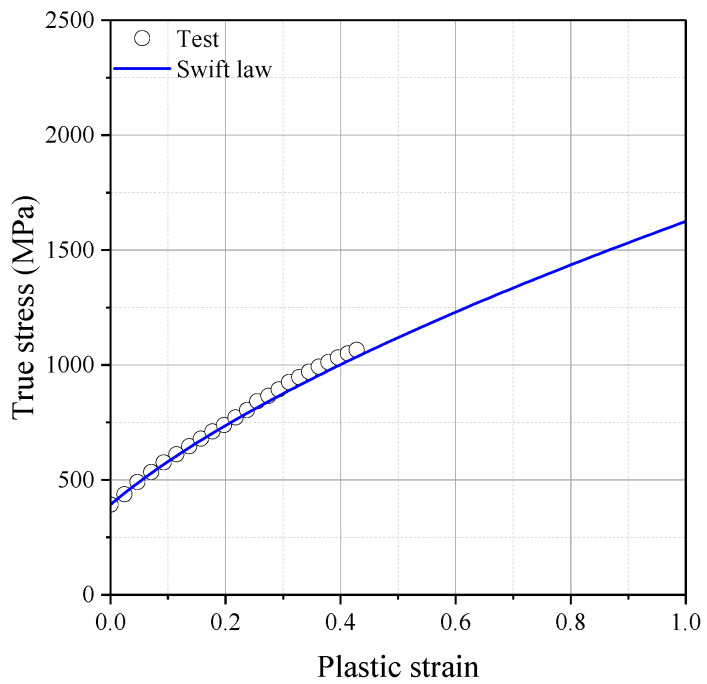
Teste data [28] and re-modeled hardening curves for reference paper.

**Figure 13 materials-17-00276-f013:**
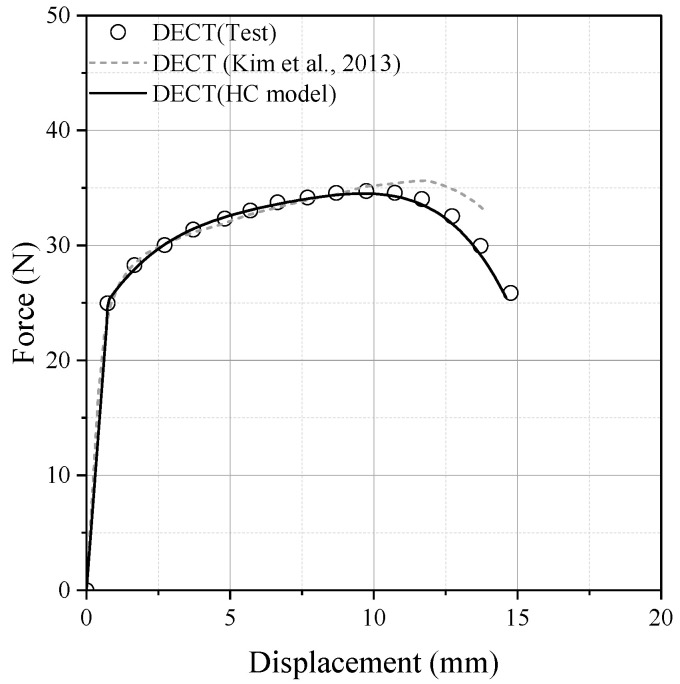
Comparison between test [28] and numerical simulations.

**Figure 14 materials-17-00276-f014:**
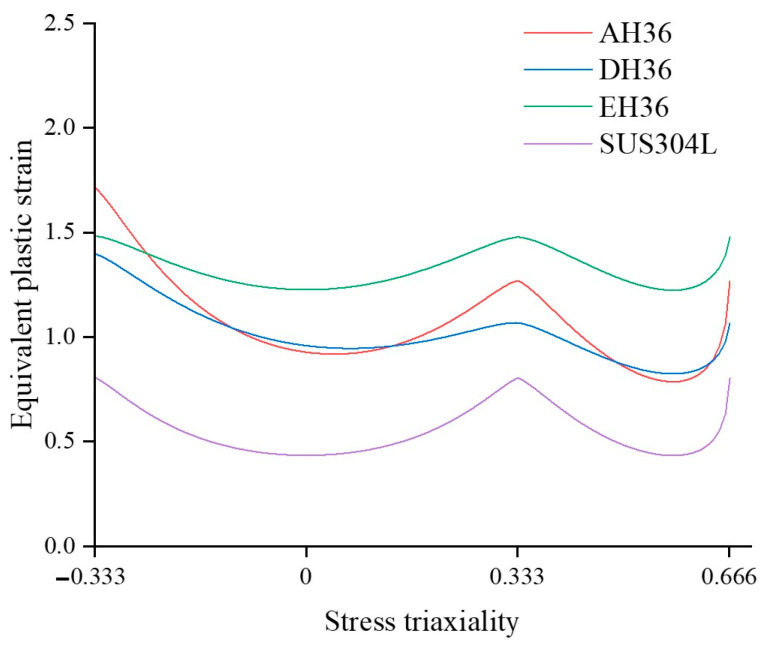
Comparison of the Hosford-Coulomb model for various steels under plane stress [25,30].

**Table 1 materials-17-00276-t001:** Chemical composition of SUS304L (%).

C	Si	Mn	P	S	Cu	Cr	Ni	Mo
0.0224	0.459	1.429	0.0321	0.0047	0.276	18.148	8.080	0.189

## Data Availability

Data are contained within the article.

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
