# Peer review of "Numerical Study on a Ductile Fracture Model in Pre-Cracked Tension Tests of SUS304L"

_materials, 2024, doi:10.3390/ma17020276_

Round 1

Reviewer 1 Report

Comments and Suggestions for Authors

The paper presented research on plasticity and ductile fracture models applicable to SUS304L steel, employing the Swift isotropic hardening law for plasticity and the Hosford-Coulomb model with a damage framework for simulating ductile fracture. Calibration and validation were performed through quasi-static tests and numerical simulations, particularly for the double-edge cracked tension test. Paper can eventually be published after minor revision according to the following remarks:

Please, indicate the novelty of your study in the introduction.

Did the tests take into account roughness or other surface parameters?

Author Response

The paper presented research on plasticity and ductile fracture models applicable to SUS304L steel, employing the Swift isotropic hardening law for plasticity and the Hosford-Coulomb model with a damage framework for simulating ductile fracture. Calibration and validation were performed through quasi-static tests and numerical simulations, particularly for the double-edge cracked tension test. Paper can eventually be published after minor revision according to the following remarks:

Please, indicate the novelty of your study in the introduction.

: I re-write the introduction part.

Did the tests take into account roughness or other surface parameters?
: In these tests, there is no contact between specimens, and surface roughness is not taken into consideration.

Reviewer 2 Report

Comments and Suggestions for Authors

Materials

ID 2741784

REVIEW

Title: “Numerical study on ductile fracture model in pre-cracked tension tests of SUS304L

Line 97 – We are talking about (DB) but Fig. 1 does not show the related specimen with dimensions as in the other examples.

Fig. 1 – Although it shows the drawing for (FB), the text does not make any reference to it, so it is not known what this acronym represents.

Eq. (1) – Who is  Ɛp  ?

Eq. (3) – Who are Ɛ0, A and n?

Line 122 – There is talk about (DB) again, but he was not explained in the previous text.

Line 129 -130 – The numerical values given must be justified. They appear without explaining how they were obtained (by calculation or chosen from tables and according to what criteria).

Fig.3 – There are data about FE models for NT, PST, and SH, but the parameters A, Ɛ0, and n are given for (DB). What is the value of the same parameters for NT, PST, and SH and why do they differ (or why would they be the same)?

Fig. 4 – How was the curve drawn? With what device? In what experimental conditions? For which of the three (NT), (PST) or (SH) is the experimental curve valid?

Eq. (5)- What does s mean? It must be explained!

Eq. (9) - It must be explained which of the axes σ1, σ2, and σ3 are valid.

Eq (15) - What does σf mean?

Line 166 – Who is Ɛf?

Fig. 6 – (DB) and (FB) are missing. If you do not refer to them, then do not mention them in the text.

Fig. 6 – How the test curves were obtained that are compared with the results obtained with ABACUS. Specify the installation and the method of drawing the curves.

Line. 180 – It is the first time that the parameters {a, b, c} appear and it is not explained what they represent in the HC model.

Line 186-187 – Where do the values of parameters a,b, and c come from?

Fig. 8 – Why the plane represented by the coordinates "Equiv. plastic strain"(OY) with (OX)-"Stress Triaxiality" in 8(a) does not correspond to 8(b). From a straight line that is given by the mentioned section, we moved to a complex line in the presented section. No section through the 3D curve shown gives such a complex curve.

Line 201 – How to upload the sample? Who ensures the rhythm of 1mm/min? Where is the charge applied? Everything is presented in a blur for the experiment.

Fig. 10 – It is not clear which part of the DECT specimen was taken as FE.

Line 220 - Same question. Where do the numerical values for A, Ɛ0, and n?

Fig.11 - How was the test curve drawn? With what device and how was it recorded? As well as in fig. 12, the same question. What does the broken line in the diagram represent and what does it mean?

Conclusion – The work presents itself as a confirmation of the HC model. It does not appear what the original contributions of the authors are in the research. It is only stated that it contributes to a better understanding of fracture predictions in pre-fractured structures.

General observation

If the pre-fractured structures were studied, shouldn't the state structure of the material be described first before the finite element analysis? How were the parts with pre-cracked approaches when discretizing? The ones presented in the paper represent an approach that does not depend on the placement of pre-cracked in the sample structure. So, although the numerical approach to pre-cracked appears in the title, they are not addressed in the paper, but an approach is made regardless of the structure. In its current form, the work is at the level of applying some known methods to a structure indifferent to the existing pre-cracked. The work cannot be published in this state.

Author Response

Line 97 – We are talking about (DB) but Fig. 1 does not show the related specimen with dimensions as in the other examples.

: Check and modify in main body.

Fig. 1 – Although it shows the drawing for (FB), the text does not make any reference to it, so it is not known what this acronym represents.

: Check and modify in main body.

Eq. (1) – Who is  Ɛp  ?

: Check and add in main body.

Eq. (3) – Who are Ɛ0, A and n?

: Check and add in main body.

Line 122 – There is talk about (DB) again, but he was not explained in the previous text.

: I standardized the terminology for specimens in the main body, changing it from "Dog-bone" to "Flat bar" (from DB to FB).

Line 129 -130 – The numerical values given must be justified. They appear without explaining how they were obtained (by calculation or chosen from tables and according to what criteria).

: This curve is obtained using MATLAB curvefitting tool with Swift equation model. I added it in main boddy.

Fig.3 – There are data about FE models for NT, PST, and SH, but the parameters A, Ɛ0, and n are given for (DB). What is the value of the same parameters for NT, PST, and SH and why do they differ (or why would they be the same)?

: In this study, the stress-strain curves before the onset of necking were obtained using standard tension tests on flat bar specimens, and the flow stress beyond the large plastic strain region is extrapolated using the hardening law. The presented hardening model parameters were verified through the numerical analysis for NT, PST and SH cases.

Fig. 4 – How was the curve drawn? With what device? In what experimental conditions? For which of the three (NT), (PST) or (SH) is the experimental curve valid?

:It is identical to the preceding statement.

Eq. (5)- What does s mean? It must be explained!

: I add the information for it in main body à They represent the combined influence of the magnitude and direction of shear stress.

Eq. (9) - It must be explained which of the axes σ1, σ2, and σ3 are valid.

: The von Mises stress is a scalar value derived from the principal stresses. These three principal stresses are considered valid in its calculation.

Eq (15) - What does σf mean?

: σf is a von Mises equivalent stress at fracture under proportional loading.

Line 166 – Who is Ɛf?

: Check and add in main body.

Fig. 6 – (DB) and (FB) are missing. If you do not refer to them, then do not mention them in the text.

: The flat bar (FB) test was exclusively utilized for calibrating the flow stress, as illustrated in Figure 4.

Fig. 6 – How the test curves were obtained that are compared with the results obtained with ABACUS. Specify the installation and the method of drawing the curves.

: Add in main body à Determining the fracture initiation displacement is challenging, especially if it occurs in-side the specimen, even when measured using a monitoring device. Fracture initiation was assumed to take place at the element where the maximum equivalent plastic strain is observed during fracture displacement. Figure 7 displays contour plots obtained from fi-nite element analysis with equivalent plastic strain. In the notch tension specimen, fracture initiation occurs at the center of the notch, while in the plain strain tension specimen, it is observed at the side of the notch. For the shear specimen, fracture occurs at the center of the notch.

Line. 180 – It is the first time that the parameters {a, b, c} appear and it is not explained what they represent in the HC model.

: Add in main body à the Hosford-Coulomb model consists of three parameters .

Line 186-187 – Where do the values of parameters a,b, and c come from?

: It is derived through the optimization process utilizing Equation 21.

Fig. 8 – Why the plane represented by the coordinates "Equiv. plastic strain"(OY) with (OX)-"Stress Triaxiality" in 8(a) does not correspond to 8(b). From a straight line that is given by the mentioned section, we moved to a complex line in the presented section. No section through the 3D curve shown gives such a complex curve.

: Figure b is under a plain stress condition. I have included a plain stress line in 3D for clarity.

Line 201 – How to upload the sample? Who ensures the rhythm of 1mm/min? Where is the charge applied? Everything is presented in a blur for the experiment.

: this test speed is presented in the reference paper

  • Kim, S.-K.; Lee, C.-S.; Kim, J.-H.; Kim, M.-H.; Lee, J.-M. Computational Evaluation of Resistance of Fracture Capacity for SUS304L of Liquefied Natural Gas Insulation System under Cryogenic Temperatures Using ABAQUS User-Defined Material Subroutine. Mater Des 2013, 50, 522–532. https://doi.org/10.1016/j.matdes.2013.03.064.

Fig. 10 – It is not clear which part of the DECT specimen was taken as FE.

: add in the main body à The finite element model was created under 1/8 symmetric conditions for length, width, and thickness.

Line 220 - Same question. Where do the numerical values for A, Ɛ0, and n?

: It is derived through the same process as illustrated in Figure 4.

Fig.11 - How was the test curve drawn? With what device and how was it recorded? As well as in fig. 12, the same question. What does the broken line in the diagram represent and what does it mean?

:It is identical to the preceding statement.

Conclusion – The work presents itself as a confirmation of the HC model. It does not appear what the original contributions of the authors are in the research. It is only stated that it contributes to a better understanding of fracture predictions in pre-fractured structures.

: re-write the conclusion

General observation

If the pre-fractured structures were studied, shouldn't the state structure of the material be described first before the finite element analysis? How were the parts with pre-cracked approaches when discretizing? The ones presented in the paper represent an approach that does not depend on the placement of pre-cracked in the sample structure. So, although the numerical approach to pre-cracked appears in the title, they are not addressed in the paper, but an approach is made regardless of the structure. In its current form, the work is at the level of applying some known methods to a structure indifferent to the existing pre-cracked. The work cannot be published in this state.

: In this study, the presented fracture model was validated through numerical analysis using a double-edge pre-cracked tension test, as described in the reference paper. The pre-crack process is not detailed in the numerical simulation within the reference paper; it is solely compared with the force level from the pre-crack (with the initial crack) to fracture. In our study, we specifically focus on specimens with an initial crack and do not consider the crack process.

Reviewer 3 Report

Comments and Suggestions for Authors

In this study, The effectiveness of a ductile fracture model in accurately predicting fracture initiation has been demonstrated. It is interesting three kinds of tension specimens with different fracture modes were designed. 

1. What is the special difference between the simulation model proposed by the author and the model of other scholars? Where is the innovation?

2. Why did the author choose these three kinds of tension specimens as research objects?

3. The author has verified the accuracy of the model when introducing three kinds of tensile samples, why does the author need to illustrate the accuracy of the model through notched samples?

4. As far as I consider, it would be better if the author paid more attention to the fracture process rather than stressing the accuracy of the model repeatedly.

Author Response

  1. What is the special difference between the simulation model proposed by the author and the model of other scholars? Where is the innovation?

: Key contributions include a focus on ductile fracture modeling, utilization of fracture models like the Swift hardening law and Hosford-Coulomb model, consideration of pre-cracked structures, material and test selection, numerical analysis with calibration, and validation through experiments. The research addresses a gap in SUS304L materials, contributing valuable insights for safety, structural assessment, and the development of reliable engineering components. The fracture characteristics of SUS304L, a material being considered for hydrogen tank usage, have not garnered as much attention from researchers as normal high-strength steel and marine structural steel. Moreover, additional exploration is required for the fracture model in extremely low temperature.

  1. Why did the author choose these three kinds of tension specimens as research objects?

: Each specimen exhibits a distinct fracture mode, as depicted in Figure 8. The commonly used shear fracture criteria in the industry assume that fracture occurs when the equivalent plastic strain reaches a critical value. However, various researchers have shown that the fracture strain varies depending on the stress state. Therefore, we conducted tests on specimens with different fracture modes to validate the model.

  1. The author has verified the accuracy of the model when introducing three kinds of tensile samples, why does the author need to illustrate the accuracy of the model through notched samples?

: Three types of tensile samples were included in the training set for the optimization process. We are of the opinion that specimens used for the test set might not be appropriate for the validation set (cross-validation). Therefore, we utilized alternative tests for the validation of the presented model.

  1. As far as I consider, it would be better if the author paid more attention to the fracture process rather than stressing the accuracy of the model repeatedly.

: In this study, the presented fracture model was validated through numerical analysis using a double-edge pre-cracked tension test, as described in the reference paper. The pre-crack process is not detailed in the numerical simulation within the reference paper; it is solely compared with the force level from the pre-crack (with the initial crack) to fracture. In our study, we specifically focus on specimens with an initial crack and do not consider the crack process.

Round 2

Reviewer 2 Report

Comments and Suggestions for Authors

With the modifications made after review, the paper deserves to be published.